# Processability of 21NiCrMo2 Steel Using the Laser Powder Bed Fusion: Selection of Process Parameters and Resulting Mechanical Properties

**DOI:** 10.3390/ma15248972

**Published:** 2022-12-15

**Authors:** Jakub Łuszczek, Lucjan Śnieżek, Krzysztof Grzelak, Janusz Kluczyński, Janusz Torzewski, Ireneusz Szachogłuchowicz, Marcin Wachowski, Marcin Karpiński

**Affiliations:** 1Institute of Robots & Machine Design, Faculty of Mechanical Engineering, Military University of Technology, 00-908 Warsaw, Poland; 2Centre of Functional Materials, Lukasiewicz Research Network—Institute of Non-Ferrous Metals, 44-100 Gliwice, Poland

**Keywords:** mechanical engineering, additive manufacturing, laser powder bed fusion, 21 NiCrMo2 steel, process parameters, post-heat treatment

## Abstract

With the development and popularization of additive manufacturing, attempts have been made to implement this technology into the production processes of machine parts, including gears. In the case of the additive manufacturing of gears, the availability of dedicated materials for this type of application is low. This paper summarizes the results of research on the implementation of 21NiCrMo2 low-alloy steel, which is conventionally used to produce gears as a feedstock in the PBF-LB/M process. The work presents research on the selection of process parameters based on porosity measurements, static tensile tests, and hardness measurements. In addition, the article includes a mathematical model based on the quadratic regression model, which allows the estimation of the percentage of voids in the material depending on the assumed values of independent variables (laser power, scanning velocity, and hatch distance). The paper includes a range of process parameters that enable the production of elements made of 21NiCrMo2 steel with a density of over 99.7%. Additionally, comparative tests were carried out on PBF-LB/M-manufactured steel (in the state after printing and the state after heat treatment) and conventionally manufactured steel in terms of its mechanical and microstructural properties. The results showed that the steel exhibited similar mechanical properties to other carburizing steels (20MnCr5 and 16MnCr5) that have been used to date in PBF-LB/M processes and it can be used as an alternative to these materials.

## 1. Introduction

Additive manufacturing (AM) (according to ISO/ASTM 52900:2021) is a process used for the joining of materials based on 3D CAD data. In contrast to subtractive methods, this technology is characterized by building parts layer-by-layer. The inclusion of these data in the standardized definition is the main advantage of AM technology. Hence, it is possible to obtain complex geometries, such as internal conformal cooling channels and lattice structures, which are difficult to produce with the use of conventional manufacturing technologies. In the manufacturing of machine parts via AM, powder bed fusion (PBF) (ISO/ASTM 52900) [1] technologies are most commonly used. It is worth noting that in numerous studies, the most popular method has involved the use of a laser energy source. This method is referred to as laser-based powder bed fusion of metals (PBF-LB/M)—ISO/ASTM 52911-1—and is popularly called laser powder bed fusion (L-PBF). The use of PBF technology in the production of machine parts must be economically justified. Many different research articles related to PBF-LB/M have discussed this issue [2,3,4,5]. The mentioned works indicate that the purchase costs of machines and materials represent the largest part of the financial expenditure in terms of production. On the one hand, Kamps et al. [6] pointed out that material expenses in the PBF-LB/M process constituted 32% of the overall cost. On the other hand, the American Gear Manufacturers Association (AGMA) [7] reported that the value was equal to 15%, dependent on the feedstock material type. In both cases, those costs are high and are included in the cost share of the powder in the overall process, as reported by Lindemann et al. [8]. The atomization process consumes most of the costs per 1 kg of powder. With the current balance of costs, the profitability of production using PBF-LB/M technology is achieved in unit production, in particular through the complex geometry of the parts produced. In order to expand the areas in which PBF-LB/M technology is applicable, it is necessary to develop new materials or implement those which are used in conventional processes, e.g., low-alloy steels.

One of the basic elements in the field of mechanical drive systems, which can be successfully produced via PBF-LB/M, is gears [9,10]. The loads acting on gears require the use of proper feedstock materials and postprocessing activities, which allow the proper strength properties to be obtained. Numerous studies have taken into account the production processes of power train components and the verification of their mechanical properties [11,12,13,14]. The main issue in these papers is the use of materials that are not typically dedicated to some exact solution. The most common commercially available metallic powders for PBF-LB/M are Ti6Al4V [14], AlSi10Mg [15], and the stainless steels 316 L [15], 420 [16], 17-4PH [13], and GP1 [17]. The number of steel grades that are intended for AM gears is limited [7]. One of the primary properties which allows the use of a given material in the PBF-LB/M process is its weldability (which is mostly dependent on its having a carbon amount below 0.3%). However, this feature can be modified by adding appropriate alloying elements. The most common conventionally used steels which have been implemented into PBF-LB/M processes are 16MnCr5 and 20MnCr5 [18,19,20], both of which are carburizing steels. According to the AGMA report [7], for this type of AM method, the following materials are in use: Pyrowear 53, Pyrowear 675, Ferrium C61, Ferrium C64, 100Cr6, M50 NIL, AISI 8620, and AISI 9310. Additionally, not all of the abovementioned materials are available in the commercial market [7]. Moreover, in the actual state of the art, some studies have been reported in which the authors used steels for quenching and tempering, i.e., 42CrMo4 [21,22], 30CrNiMo8 [23], 24CrNiMo [24], and HSLA-100 [25].

For the use of construction steels (including carburizing, quenching, and tempering) in the AM process, it is necessary to provide a properly prepared powder and the detailed development of process parameters. In the case of 16MnCr5 carburizing steel, Schmitt et al. [20,26] proved that it is possible to use AM for the production of gears without any internal cracks and imperfections and to maintain a high material density above 99.5% by means of an energy density above 100 J/mm^3^ (for 99.94% it is necessary to use 108.3 J/mm^3^). With such high densities, the main defect was regular shape porosity, described as gas porosity. The authors highlighted that when using the proper energy density value with different combinations of process parameters (laser power, exposure velocity, and hatching distance) they observed changes in porosity and strength properties. The ultimate tensile strength (UTS) of the obtained parts was equal to 1050 MPa and had the same value as yield strength (YS), which indicates that there was no plastic deformation and thus also no work hardening. Before the heat treatment, the conventionally manufactured material was characterized by UTS = 715 MPa and YS = 591 MPa, which represent lower values than the AM material, at 32% and 43.7%, respectively. The heat treatment (stress relief annealing) conditions for AM parts were the same as those used for the conventionally made materials. This approach resulted in a drop in the mechanical properties (UTSATH = 730 MPa, YSATH = 658 MPa, hardness from 330 HV10 to 235 HV10); additionally, it led to the creation of a fine microstructure and the elimination of the layered structure of the material. Based on the authors’ work [20,26], such changes indicate a gain in the recrystallization temperature. Additionally, microscopic investigation revealed structural heterogeneities as “white areas” after nital etching. Kamps et al. [27] and Scheitler et al. [28] proved the existence of this kind of structural anomaly in additively manufactured steel parts. Kamps et al. [27] revealed that the chemical composition of the “white area” structure is similar to that of the base material. Neither of the mentioned authors described the formation mechanism of this structure. Yang and Sisson [19] additively manufactured samples with the use of 20MnCr5 carburizing steel, employing PBF-LB/M technology. However, neither production parameters nor porosity values were given. Their research focused on the influence of heat and chemical treatments on the hardness of AM parts. The measurements were conducted using two perpendicular cross-sections of the aforementioned samples. In the as-built parts, the hardness value was equal to 287 HV5 in the XY and YZ planes. At the same time, the measured value of the parent material was 189 HV5. The differences between the mentioned material types (the AM and parent material) were mostly caused by the different forms of microstructures. The parent material exhibited a ferritic-pearlitic structure, whereas the AM parts displayed a martensitic structure. The only parameters reported for the production of 20MnCr5 steel using the PBF-LB/M technique were provided in the work of Robbato et al. [18]—106 J/mm^3^ (counted on base given process parameters). However, the authors of that study did not report the porosity value obtained. Based on the works related to 16MnCr5 steel, it can be concluded that the density values were similar in both cases.

In the case of quenching and tempering steel, Zumofen et al. [23] used 30CrNiMo8 in the PBF-LB/M process. The authors used the energy density at the level of 83 J/mm^3^, which allowed them to obtain a density at the level of 99.76%. The remaining voids in the material structure were characterized by a spherical shape, which is a typical indicator of gas porosity. There were no cracks in the structure of the material. The UTS and YS values of the printed material in the quenched and tempered state were higher than those in the as-built state and were comparable with those of the parent material after similar heat treatment. 42CrMo2 steel was used in the PBF-LB/M process in several works [21,22]. Damon et al. [21] achieved a porosity of 0.3% using an energy density of 85 J/mm^3^. No cracks were registered in the structure of the material. As in the case of 30CrNiMo8 steel, the authors obtained higher strength parameters (UTS and YS) of the steel after quenching and tempering treatment than in the as-built state. Moreover, for this material in the as-built condition, the microstructure was also characterized by an acicular, fine-grained martensitic structure.

Based on the literature review, it can be concluded that the number of low-alloy steels used in additive manufacturing is still meager. In the case of steel for carburizing, it is necessary to use a higher energy density (above 100 J/mm^3^) in order to obtain a density above 99.7% than in the case of steel for quenching and tempering (around 85 J/mm^3^). In both cases, the materials in the as-built state are characterized by higher hardness and strength than the parent material. This is due to their acicular, fine-grained martensitic, or martensitic-bainitic microstructure. The change in the properties of the additive manufacturing material depends on the heat treatment performed. The particular deficiencies in the broad description of the processability of steels using PBF-LB/M are related to carburizing steels. The literature lacks a description of steels that are alternatives to the widely used 16MnCr5 or 20MnCr5. In this study, 21NiCrMo2 steel was used in order to investigate its processability with the use of PBF-LB/M, as well as the mechanical properties obtained through this process. This will provide an answer as to whether 21NiCrMo2 steel can be used as an alternative to the typical carburizing steels used to date.

## 2. Materials and Methods

The material taken into account in this research was the 21NiCrMo2 alloy. Such steel is traditionally used for the production of machinery equipment parts, i.e., gears, shafts, etc. Because of the limited availability of such material in the form of dedicated powder for PBF-LB/M processes, conventionally made 21NiCrMo2 steel bars were subjected to the gas atomization (GA) at the Institute of Non-Ferrous Metals (Gliwice, Poland). The LD Vacuum Technologies GmbH atomizer (Hanau, Germany) was used for powder production. The GA process parameters are shown in Table 1.

The powder obtained during GA was sieved using a Retsch AS200 sieve shaker (Microtrac Retsch GmbH, Haan, Germany) which allowed us to separate the proper powder fraction for the PBF-LB/M process (20–63 µm). Most powder grains (indicated by red arrows) were characterized by spherical shapes with some satellites (indicated by green arrows), which is visible in Figure 1.

In further analyses, the material’s chemical composition was investigated with the use of a scanning electron microscope (SEM) JEOL JEM-1230 (Jeol Ltd., Tokyo, Japan) equipped with an energy-dispersive spectroscopy (EDS) module. Spot measurements were made on the surfaces of the powder particles. Table 2 contains the EDS measurement results. Due to the fact that the measurements of the carbon content using the EDS method were flawed due to a large error, this value was not included in Table 2. Moreover, the authors did not have any other equipment at their disposal to conduct this type of research. During that process, there were not any material heterogeneities registered.

### 2.1. Laser Powder Bed Fusion (PBF-LB/M)

SLM 125HL (SLM Solutions GmbH, Lubeck, Germany) was used for the AM samples. The device was equipped with a 400 W single Ytterbium-fiber-laser source (wavelength 1080 nm) and a maximum scanning velocity of 10 m/s. The maximum build volume is equal to 125 × 125 × 125 mm. The range of possible layer thicknesses was 20–75 µm. The substrate plate could be heated up to 200 °C, and the AM process was performed in an argon atmosphere (in which the amount of oxygen was lower than 0.3%).

The first stage of the study involved the development of process parameters based on the porosity analysis of the AM cubic samples. Regarding the limited availability of information on the process parameters for 21NiCrMo2 steel, the default settings for H13 tool steel were used as base parameters (laser power *P_L_* = 225 W, scanning velocity *v_s_* = 600 mm/s, hatch distance *d_H_* = 0.120 mm). To properly prepare for the parameter development stage, 57 different parameter groups were considered. The exact values of process parameters were tested in the following ranges: laser power *P_L_* ranging from 160 W to 240 W (not using the full laser power due to its tendency to generate cracks in the steel structure), scanning velocity v_s_ ranging from 600 to 1100 mm/s, and hatching distance *d_H_* ranging from 0.070 mm to 0.120 mm. The layer thickness *l_t_* was kept at the same level and was equal to 0.03 mm. The platform heating was set at a value of 190 °C. As a representative parameter (dependent on *P_L_*, *v_s_*, *d_H_*, and *t_L_*) energy density *E_v_* can be described by means of the following Equation (1):(1)$$E_{v}=\frac{P_{L}}{v_{s}\cdot d_{H}\cdot t_{L}}\;\left[\frac{J}{mm^{3}}\right]$$

AM samples for the porosity analysis had the form of cubes with an edge length equal to 10 mm (one sample for each parameter group). The porosity measurements were performed utilizing the Keyence VHX 7000 optical microscope (Keyence, Osaka, Japan) in both representative planes: XY(P_ρXY_)—parallel to the substrate plate surface, and YZ (P_ρYZ_)—perpendicular to the substrate plate surface (Figure 2). The samples were cut using wire electrical discharge machining (WEDM) and mounted in resin for further microscopical investigation. All samples were ground using abrasive papers with a gradation from 320 to 2400 and polished using 1 µm diamond paste. As a representative porosity value, the average value was taken from P_ρXY_ and P_ρYZ_ (three different measurements for each plane).

The maximum acceptable porosity value in the entire area of measurement in cross-sections was equal to 0.3%. Additionally, to improve the development of process parameters, Design of Experiment (DOE) analysis using Statistica software 13.1 (TIBICO Software Inc., Palo Alto, Santa Clara, CA, USA) was used. For the description of mathematical porosity values, the quadratic area regression model was used. This selection was made because of the possibility of combining features of multinomial regression and fraction factorial regression models. Hence, it allowed the consideration of three independent variables and their mutual interaction. Adegok et al. [29] suggested such an approach in their work. The general form of the quadratic regression is shown in Equation (2):(2)$$y=\beta_{0}+\beta_{1}x_{1}+\beta_{2}x_{2}+\beta_{3}x_{3}+\beta_{11}x_{1}^{2}+\beta_{22}x_{2}^{2}+\beta_{33}x_{3}^{2}+\beta_{12}x_{1}x_{2}+\beta_{13}x_{1}x_{3}+\beta_{23}x_{2}x_{3}+\in$$

In Equation (2), y is a dependent variable (the estimated porosity value) and *x*_1_, *x*_2_, and *x*_3_ are independent variables, which can be described as follows:*x*_1_—laser power;*x*_2_—scanning velocity; and*x*_3_—hatching distance.

β_m_ and β_mn_ (for m = 1, 2, 3; *n* = 1, 2, 3) are regression coefficients, and ε is the modeling residual or error. The values of the regression coefficients were calculated with the use of the method of least squares. The calculations were divided into two parts. The first stage was dedicated to the creation of the model, based on the first porosity measurements (based on two experiments) of 27 different samples. The second stage was based on the development of further parameter combinations using the 3^3^ full factorial designs (three factors: laser power, exposure speed, and hatching distance were varied at three levels for each factor). Such an approach allowed us to supplement the statistical model with the obtained results.

Additionally, the R^2^ and *p* values were calculated. The R^2^ coefficient defines how the statistical model and its predictors describe the variability of a referred parameter. The *p* value is the cumulative probability of drawing a sample as extreme as or more extreme than the observed one, assuming that the null hypothesis is true. The *p* coefficient was estimated by constructing an analysis of variance (ANOVA) table, and this was related to statistical tests. Statistical significance was set at *p* < 0.05. The validation of the obtained PBF-LB/M process parameter groups with the statistical analysis results was possible via the experimental study of the microscopic observations and porosity measurements. As a result, five process parameter groups were chosen for further research (microstructure investigation, hardness testing, and tensile tests).

### 2.2. Microstructure and Tensile Analysis

As-built PBF-LB/M samples and parent material parts were considered for the microstructural and tensile analyses. Because of the presence of high-temperature gradients, after the PBF-LB/M processing, all manufactured parts were subjected to stress relief annealing [30]. The lack of this operation caused deformation of samples during their separation from the substrate plate via WEDM. The heat treatment process was undertaken in a Nabertherm N11/H furnace (Nabertherm GmbH, Lilienthal, Germany). The conventionally made material was examined after normalization. All the details of the annealing process are shown in Table 3 [31]. Temperature and time values were taken from the heat treatment of conventionally made 21NiCrMo2 steel.

Dog-bone tensile samples were designed based on the ASTM E8 standard, and all the samples were oriented as shown in Figure 2b. This is the most favorable position for the specimens in terms of the strength of the additive manufacturing material when subjected to static tensile tests. Tensile tests were conducted on the INSTRON 8802 MTL (Instron, Norwood, MA, USA) testing machine in accordance with the PN-EN ISO 6892-1:2010 standard. Using each selected group of parameters, 5 samples were produced and tested using a tensile test. In the case of hardness testing, all measurements were made by means of a Struers DuraScan 70 system (Struers GmbH, Kopenhagen, Denmark) in accordance with the PN-EN ISO 6507-1:2007 standard. Hardness tests were conducted on the same samples, which were dedicated to the investigation of porosity. To illustrate the methodology presented here, Figure 2a shows a flowchart that briefly summarizes Section 2.

## 3. Results

### 3.1. Process Parameters and Prediction Model

The first stage of process parameter development included the AM of 27 samples. All data and predicted porosity results are included in Table A1. The modification ranges of the parameters were as follows:*P_L_* from 160 to 240 W;*v_s_* from 600 to 1100 mm/s; and*h_d_* from 0.070 to 0.110 mm.

The groups of parameters in which the requirement $\bar{P_{\rho }}$ < 0.3% was achieved were characterized by energy densities ranging from 95.2 to 121.2 J/mm^3^. Additionally, we ben observed that the measured porosity values were different despite reaching similar values of energy density in the given groups. The 1.25, 1.26, and 1.27 parameter groups (Table A1) taken from the available literature [23,27] did not achieve the required porosity regime. Equation (1) was used to estimate porosity values for these cases. As variables, three parameters were considered: *x*_1_—laser power, *x*_2_—exposure speed, and *x*_3_—hatching distance. Using the method of least squares, the coefficients β_m_ and β_mn_ (for m = 1, 2, 3; *n* = 1, 2, 3) were established. Additionally, by constructing the analysis of the variance table, it was possible to determine the statistical significance and create Equation (3):(3)$$y=52.149000-0.018000x_{1}-0.067000x_{2}-525.55000x_{3}+0.00032000x_{1}^{2}+0.000031x_{2}^{2}\;+1499.389000x_{3}^{2}-0.000099x_{1}x_{2}-0.475000x_{1}x_{3}+0.449000x_{2}x_{3}+0.580000$$

According to a different number of considered cases for *x*_1_, *x*_2_, and *x*_3_ variables, the statistical significance was valid for only one part of the equation (exposure speed, *p* = 0.015). The value of the R^2^ coefficient for this exact case was equal to 0.83. Based on the established minimum of a function (3), it was possible to estimate the parameter groups, ensuring the same density as the parent material. Additionally, a determinizing estimation of the process parameters groups was made. This approach allowed us to obtain one additional group of parameters, with which the theoretical porosity value of the produced sample would be close to that produced using the extreme of function (3). The obtained process parameter groups are shown in Table 4.

Before validating the obtained parameter groups, we decided to create 3^3^ full factorial designs to distribute the measurement points equally in the considered space, which was restricted by the given variables (x_1_, x_2_, and x_3_). Based on the obtained results (included in Table A1), it was possible to identify three process parameter value for each variable. The selection was made according to the lowest measured porosity achieved using an exact parameter group. In the case of the hatching distance, as an additional third variable, a value of 0.120 mm was taken into account.

This approach allowed for the generation of 27 different parameter groups (Table A2). Moreover, based on Equation (3), two additional groups of parameters 2.28 and 2.29 (Table 4) were estimated. Samples manufactured through the selected parameter groups were examined using the same procedure as was performed in the first part of the experiment (Section 2.1). Ten out of a total of 29 parameter groups achieved the porosity condition ($\bar{P_{\rho }}$ < 0.3%). Subsequently, 2.29 groups (extreme of function (4)) indicated the minimum value of porosity ($\bar{P_{\rho 2.29}}$ = 0.08%) among all the tested samples.

Another level in the selection of the process parameters in the initial statistical model was supplemented by the examination of porosity measurements. This influenced the changes in the regression and *p* coefficient values (Table 5). Based on the *p*-factor, the most significant factors in the equation among the variables were the scanning velocity and the two combinations of velocity with laser power, as well as the hatch distance.

The R^2^ value equal to 0.83 did not change significantly compared to the initial value designated in the first part of this study (despite the increase in the statistical significance coefficient number). The final form of the statistical model can be described by the following Equation (4):(4)$$y=29.829000-0.011000x_{1}-0.041000x_{2}-294.833000x_{3}+0.000140x_{1}^{2}+0.000022x_{2}^{2}\;+570.583000x_{3}^{2}-0.000078x_{1}x_{2}-0.007000x_{1}x_{3}+0.278000x_{2}x_{3}+0.460000$$

The extreme of the function (4) has been considered in Table A2 as parameter group 2.30. However, the validation indicated a lack of repeatability in the results. This phenomenon was related to the adoption of a hatching distance that was too small, which caused overlapping of the exposure lines and, as a result, increased the number of voids. Hence, it was essential to consider all technical aspects related to the AM process manually because the statistical model did not take into account such limitations. Finally, based on the all obtained results, it was possible to identify five process parameter groups that were used in the further analysis:Group “2.3”—the highest value of the energy density;Groups “2.11” and “2.12”—the most repeatability after validation; andGroups “2.29” and “2.30”—the extreme function values in statistical models (3) and (4), respectively.

Statistica software was used to express the statistical model’s answers as a graphical image (only when a constant value was assumed for one of the variables). The answer surfaces generated when a constant value was used for the hatching distance parameter are shown in Figure 3.

Indicating the model answer in such a way allows one to observe the range of the process parameters in which the porosity of the AM parts would be lower than 0.3%—the so-called “technological window”. Such an estimation allows one to characterize the given material from the point of view of density without the need to conduct the whole spectrum of this kind of research. Additionally, five groups of process parameters selected for further research were shown on the answer surfaces (Figure 3). Process parameter groups 2.11 and 2.29 were located in the areas in which the statistical model estimated porosity values lower than 0.3%. Process parameter groups 2.3 and 2.12 were located close to the areas mentioned earlier.

Figure 4 shows a chart of the porosity values as a function of energy density. It can be observed that the growth of the energy density positively affected the reduction of voids in the material volume. It can also be observed that changes in the pores’ shapes were related to the increase in energy density (this is shown in the figures located in the chart in Figure 4). The use of low-energy density values caused the formation of irregular void shapes, which could be connected to the “lack of fusion” phenomenon [20]. When the energy density value was greater than 80 J/mm^3^, the share of spherical shapes (caused by gas porosity) and irregular shapes (caused by lack of fusion) of voids could be observed. After exceeding 100 J/mm^3^, the voids mainly exhibited gas porosity characteristics. For a similar value of energy density, different authors observed similar defects in the structure of the material [20,26,27]. No cracks were found in the microstructure. The point at which the line denoting the acceptable value of the porosity crossed the trend line (determined by measurement) indicated a value of energy density equal to 104 J/mm^3^. This is an empirically determined value that should be used in the AM of 21NiCrMo2 steel to achieve the condition $\bar{P_{\rho }}$ < 0.3%. For the experimentally indicated energy density (104 J/mm^3^), the estimated porosity values with the use of Equation (4) were as follows:Group 2.15: (103.9 J/mm^3^)—0.29%;Group 2.5: (104.8 J/mm^3^)—0.17%.

**Figure 4 materials-15-08972-f004:**
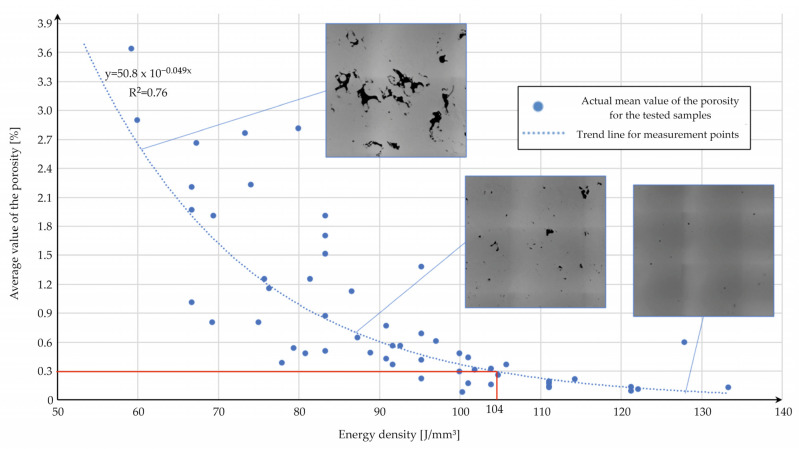
Porosity values for AM samples as a function of the energy density observed during the microscopical investigation.

These results led us to conclude that the statistical model created in this study enables us to estimate the porosity values of the AM parts made of 21NiCrMo2 steel.

### 3.2. Microstructure and Chemical Composition

PBF-LB/M and conventionally made samples were subjected to microstructural analyses (in both cross-sections). Images of the registered microstructures are shown in Figure 5. The parent material (Figure 5a) was characterized by a typical ferritic-pearlitic structure [32]. In the case of the PBF-LB/M samples, the microstructure was standard for this kind of AM process. The XY (Figure 5b) plane of the AM samples revealed exposure paths, whereas in the YZ plane, a layered structure of the deposited and melted material was visible. In addition, in Figure 5b,c, examples of porosity present in the structure are marked with green arrows. Because of the diverse cooling rates, there were observable differences in particular exposure lines, which led to a visible differentiation of the obtained microstructure. The main reason for this phenomenon is that the heat conductivity from the melt pools’ outlines was more significant than that from their center, as proven by Schmitt et al. [33]. The overall microstructure showed similarity to a martensitic-bainite structure. Because of the presence of high-temperature gradients, additional heat treatment by means of stress relief was carried out (following the guidelines shown in Table 3). Hence, we registered the loss of the layered structure and made the microstructure finer but it still exhibited the form of an acicular microstructure (Figure 5d,e). These effects are typical of recrystallization annealing. The authors of [20] also demonstrated a similar phenomenon for materials produced using PBF-LB/M after the same type of heat treatment. This effect may be due to the presence of a large number of microstructural defects in the structure in its as-built state, which may lower the temperature required for the complete rebuilding of the microstructure of the material.

Additionally, as shown in Figure 5f, in a part of the microstructure we observed a heterogeneous micro-area of concentrated alloy elements. This phenomenon was observed in both analyzed cross-sections (XY and YZ, marked with red squares and with arrows in Figure 5b,c), independently of the process parameter groups used. The dimensions of those structural heterogeneities were close to the width of a particular exposure path and a depth of 1–3 layers. According to the available research [20,27,28], such inclusions are formed during the melting process, and their chemical composition is similar to the rest of the material. In Figure 6, an EDS element distribution is identified in the mentioned micro-area.

The main alloy elements which were concentrated in the localized micro-areas were chrome, nickel, and molybdenum. The significant value of the standard deviation of the chemical composition analysis (Figure 7) indicates that a meaningful diversity of alloy elements was shared in these structures. The natural after-effect of this phenomenon was a reduction of the share of iron (Figure 6c). There is a need to conduct more detailed analyses of these phenomena in order to identify the reason for the formation of those heterogeneities.

### 3.3. Tensile Testing and Hardness Measurements

Dog-bone-shaped samples were made using the SLM 125HL device. The sample’s orientation was selected according to Figure 2. The orientation was limited to one position only, because from the point of view of material strength, this was the most advantageous position. Some specimens were chosen to undergo stress-relief heat treatment directly after the PBF-LB/M process (without the removal of samples from the substrate plate). That kind of approach allowed us to avoid the potential deformation of the manufactured parts. Conventionally made, PBF-LB/M-produced (made using the five selected parameter groups), and heat-treated PBF-LB/M samples were tested. The tensile test results are shown in Figure 8. The porosity of samples subjected to static tensile tests was also measures and it did not differ from the values obtained in the case of cubic samples.

The PBF-LB/M-made samples, in the as-built state, were characterized by a lack of a visible yield point, independent of the process parameters used. Generally, the material in the as-built state was characterized by the lowest deformation and the highest strength, which was directly related to the martensitic-bainitic microstructure of the material. The visible difference between each parameter group was registered in the strain values during the fracturing of the samples (in the as-built state). Parts obtained with the use of parameter groups 2.30 and 2.29 were characterized by smaller strain values than those of the counterparts made with the use of groups 2.3, 2.11, and 2.12. Additional heat treatment caused a change in the tensile testing curve. Through this post-processing step, it was possible to obtain a visible yield point, but at the same time, there was a significant slope in terms of the registered UTS. Excluding the samples from group 2.30, a proportional increase in the strain of the PBF-LB/M samples was visible. Additionally, parts obtained employing the parameters from group 2.30 were characterized by the highest porosity (0.54%—Table A2), which directly affected the decrease in the total strain during tensile testing. This visibly affected the total strain values, whereas the change in UTS was negligible. The parent materials in the normalized conditions were characterized by a lower UTS and a higher strain than as-built PBF-LB/M parts. This phenomenon was strictly related to microstructural material conditions. In Table 6, we present the compiled tensile test results obtained for all tested parts.

The tensile strength of the PBF-LB/M samples was close to 1012 MPa, whereas the yield point assumed values in the range of 925–970 MPa. A slight difference between given process parameter groups was also maintained in the case of samples subjected to additional heat treatment. This indicated a minor deformation strengthening effect, which is a negative phenomenon in parts dedicated to machine design. Unlike the samples obtained using parameter group 2.30, a relatively low strain value was registered during the tensile testing of parts from group 2.29. This encourages us to constantly verify the strength properties. Additional heat treatment causes a decrease in the UTS and YS to the average values of 743 MPa and 698 MPa, respectively, and an increase in the total strain to an average of 17.3% (excluding group 2.30). Under the same conditions, the 16MnCr5 materials exhibited UTS = 730 MPa and YS = 658 MPa [26].

For the hardness testing, the HV1 Vickers methodology was used. A selection of the HV1 was strictly related to the minimal tip spot hole area. The registered hardness values are shown in Table 7.

The as-built material was characterized by an average hardness of 331–357 HV1, and this was independent of the process parameter group used and the energy density used (100.3–133.3 J/mm^3^). The maximal difference between measured values in both tested cross-sections (XY and YZ) was equal to 12 HV1, which could prove a lack of anisotropy in the case of this exact material property. The fact that the highest hardness of the material was observed in the as-built state was directly caused by the presence of a fine-grained martensitic-bainite structure in contrast to the ferritic-pearlitic structure of the parent material. A decrease in the hardness was registered after additional heat treatment, with the value of 260 HV1. This was the effect of microstructural changes, which also confirms the occurrence of phenomena related to recrystallization. In both as-built and AHT conditions, PBF-LB/M material was characterized by a higher hardness value than that of the parent material, which is typical in such comparisons. The material in both states was characterized by a higher hardness than PBF-LB/M 20MnCr5 and 16MnCr5 [19,20], and this may be due to the inclusion of elements which improve hardenability in 21NiCrMo2 steel.

## 4. Conclusions

The use of 21NiCrMo2 steel in the AM process represents a significant improvement in the number of available materials from the low-alloy steel group. The research presented here proves the possibility of using such materials in PBF-LB/M processes. The developed method of process parameter selection allowed us to identify an exact range which ensured that we could reach a very high parts density, equal to 99.7%. An essential aspect of this research is the possibility of implementing the developed mathematical model in relation to other materials dedicated to the PBF-LB/M process. Through the consideration of the obtained results, we drew the following conclusions.

The developed mathematical model allowed us to estimate the range of PBF-LB/M process parameters (P_L_, v_s_, h_d_) for 21NiCrMo2 steel that can be used to obtain a material with a porosity of less than 0.3%. Moreover, this approach can be implemented for other materials when selecting the parameters of the PBF-LB/M process.We empirically determined an energy density value which should be used in the PBF-LB/M process to obtain a porosity lower than 0.3%. This should be equal to 104 J/mm^3^ or higher. At the same time, this value was the closest to the result obtained utilizing the mathematical model (4).This value is comparable with the energy density values for other carburizing steels processed via PBF-LB/M (16MnCr5, 20MnCr5) and is higher than those used for quenching and tempering steel (30CrNiMo8, 42CrMo2).A microstructural investigation allowed us to observe a share of micro-areas with alloy-element concentrations (Cr, Ni, Mo). Comparing this with the results of other studies, it can be stated that this is a phenomenon related to the low-alloy carburizing steels used in SLM technology. However, the mechanism of their formation requires deeper analysis.The stress-relieving annealing temperature for the parent material produced recrystallization mechanisms in the incrementally produced material. This is evidence that for low-alloy steels, the temperatures of heat treatments should be modified in relation to those used for the parent material.Samples obtained during the PBF-LB/M process with the use of selected parameter groups (energy densities between 100.3 and 133.3 J/mm^3^) in as-built conditions revealed the lack of a significant yield point, an increased UTS level (on average 40% higher), and a decreased strain value (in average 24% lower) in comparison to those of the parent material.Additional heat treatment in the form of stress relief caused changes in tensile testing curves (the appearance of the yield point); it also caused a reduction in the UTS from 1012 MPa to 743 MPa and an increase in total strain to 17.2% (excluding process parameter group 2.30).The HV1 hardness of as-built samples assumed values in a range of 331–357 HV1, and it was higher by 140 HV1 compared to the parent material in normalized conditions (due to the presence of a martensitic-bainitic structure for 21NiCrMo2 in the as-built state). Additional heat treatment of the PBF-LB/M as-built samples caused a decrease in hardness equal to 60 HV1.The obtained test results confirmed the comparable or higher strength and hardness of 21NiCrMo2 steel compared to other carburizing steels in the PBF-LB/M area. This clearly confirms the possibility of using 21NiCrMo2 steel as an alternative to 20MnCr5 or 16MnCr5 steel in the discussed scope of research.

Further research will be focused on the influence of thermochemical treatment (carburizing hardening, and tempering) on the microstructure, as well as mechanical and fatigue properties. This is a crucial step in order to enable more advanced applied research in this field.

## Figures and Tables

**Figure 1 materials-15-08972-f001:**
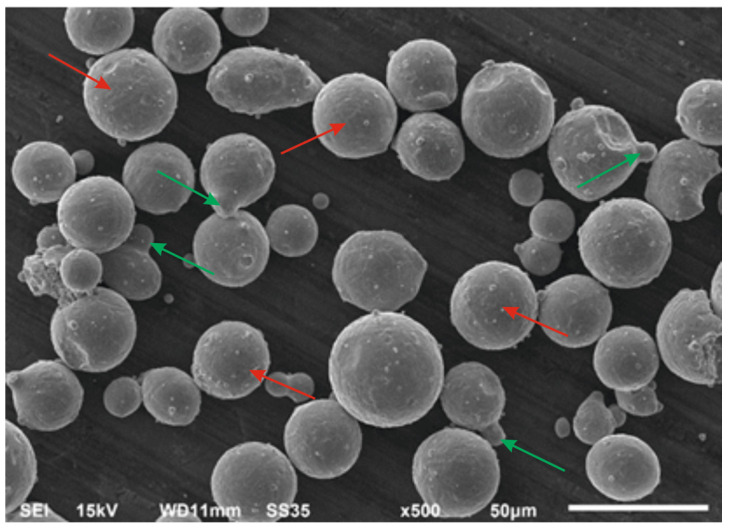
21NiCrMo2 steel powder particles.

**Figure 2 materials-15-08972-f002:**
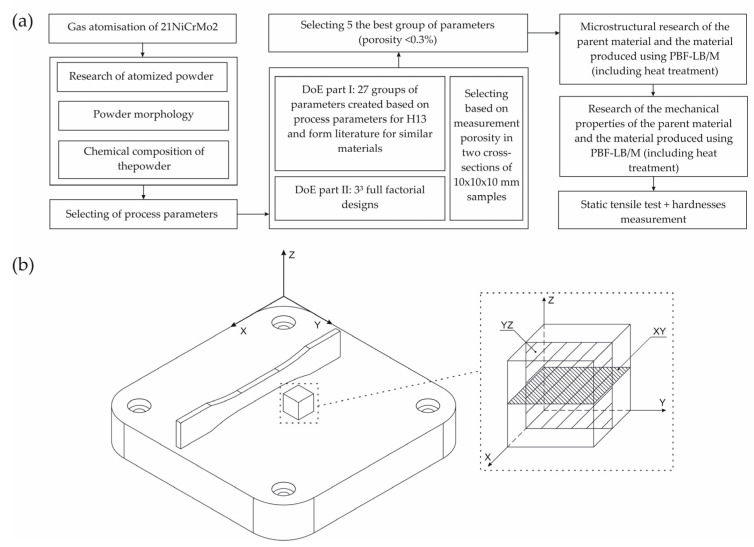
(**a**) Flowchart of the adopted research methodology and (**b**) the orientation of samples orientaton in the substrate plate of SLM 125HL (Z—the direction of the layers’ deposition).

**Figure 3 materials-15-08972-f003:**
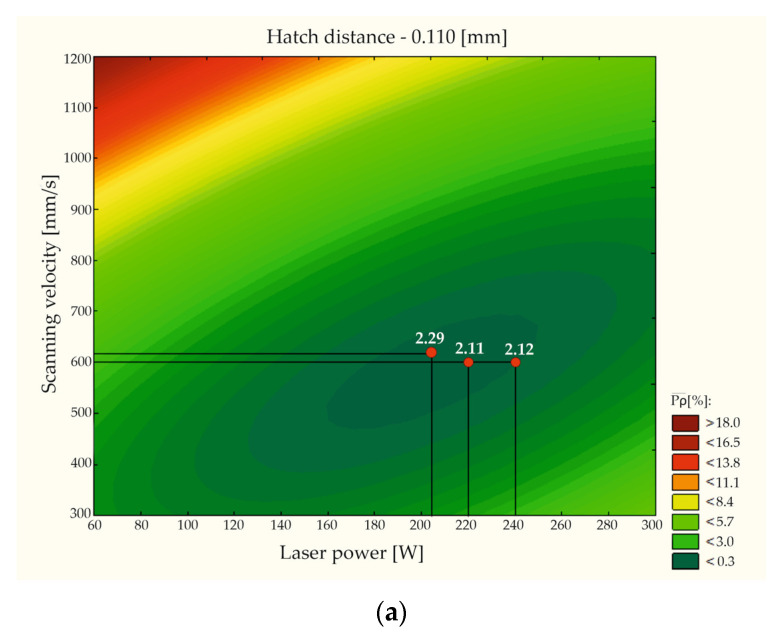
Answer surfaces generated for the constant value of the following hatching distance parameters: (**a**) h_d_ = 0.0110 mm and (**b**) h_d_ = 0.0100 mm.

**Figure 5 materials-15-08972-f005:**
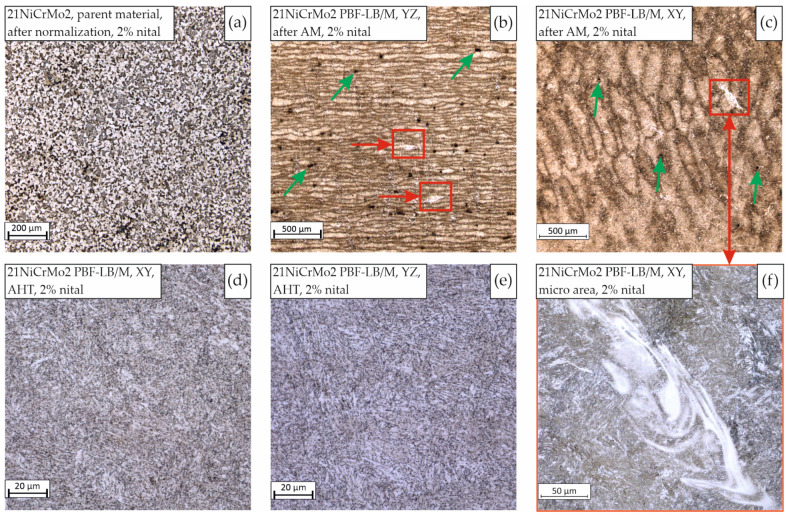
21NiCrMo2 microstructure images, 2% nital etched, obtained conventionally (**a**) and via the PBF-LB/M process in both planes: XY (**b**) and YZ (**c**); after being subjected to heat treatment (AHT) and PBF-LB/M—XY (**d**), and YZ (**e**) with the visible heterogeneous micro-area of concentrated alloy elements (**f**).

**Figure 6 materials-15-08972-f006:**
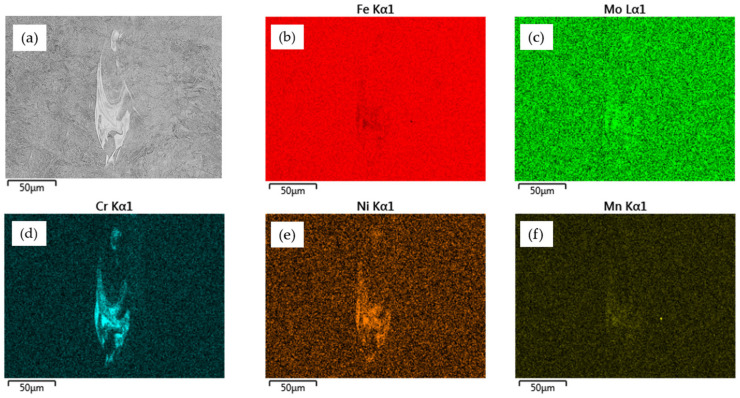
EDS element distribution identified in the 21NiCrMo2 steel area of structural heterogeneity: (**a**) micro-area image, (**b**) share of iron, (**c**) share of molybdenum, (**d**) share of chrome, (**e**) share of nickel, and (**f**) share of manganese.

**Figure 7 materials-15-08972-f007:**
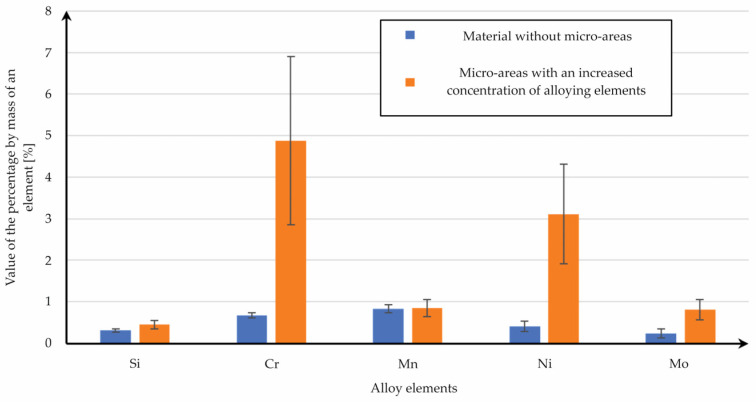
A bar chart of the exact alloy element percentages in the heterogeneous micro-area.

**Figure 8 materials-15-08972-f008:**
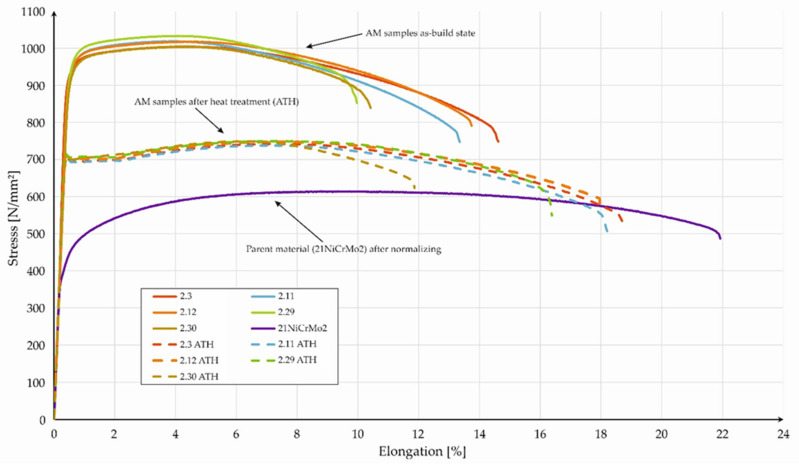
Stress-strain curves of the PBF-LB/M samples (as-built and after heat-treatment—AHT) compared with the parent materials (after normalization).

**Table 1 materials-15-08972-t001:** Process parameters used in the GA.

GA Temperature (°C)	Nozzle Diameter (mm)	Material Delivery System	Spraying Pressure (Bar)	Gas Type	Feedstock Material Type
1630	6.5	Gravitational	32	Argon	Steel bar

**Table 2 materials-15-08972-t002:** The chemical composition of GA 21NiCrMo2 steel powder.

Element	Average Measurement (wt. %)	Standard Deviation (wt. %)	Parent Material (ISO) (wt. %)
C	nd	nd	0.17–0.23
Si	0.31	0.04	<0.40
Cr	0.64	0.06	0.35–0.65
Mn	0.83	0.09	0.60–0.95
Ni	0.41	0.13	0.40–0.70
Mo	0.24	0.11	0.15–0.25
Fe	97.54	0,21	Balanced

**Table 3 materials-15-08972-t003:** 21NiCrMo2 steel heat treatment conditions [31].

Heat Treatment Type	Conditions
Stress relief annealing	Heating in the furnace until reaching 650 °C Annealing at 650 °C for 1.5 h Furnace cooling
Normalizing	Annealing in 930 °C for 4 h Air cooling

**Table 4 materials-15-08972-t004:** Process parameter values were obtained by following the statistical model based on the results of the first experiments.

Group	Variables	Condition
Extreme (minimum) of the function (3)	Laser power (W)	205
	Scanning velocity (mm/s)	619
	Hatch distance (mm)	0.110
Deterministically estimated group	Laser power (W)	212
	Scanning velocity (mm/s)	672
	Hatch distance (mm)	0.108

**Table 5 materials-15-08972-t005:** Regression and *p* coefficient values.

Coefficient Name	Coefficient Value	Value of *p* Coefficient
β_0_	29.829000	0.152869
β_1_	−0.011000	0.882832
β_11_	0.000140	0.170741
β_2_	−0.041000	0.028687
β_22_	0.000022	0.000049
β_3_	−294.833000	0.126690
β_33_	570.583000	0.337300
β_12_	−0.000078	0.003678
β_13_	−0.007000	0.988815
β_23_	0.278000	0.005153

**Table 6 materials-15-08972-t006:** Tensile test results for the parent and PBF-LB/M materials, Avr—average value, SD—standard deviation, AHT—after heat treatment.

As-Built						
**Parameter group**	Avr. UTS (MPa)	SD—Avr. UTS (MPa)	Avr. YS (MPa)	SD—Avr. YS (MPa)	Avr. elong (%)	SD—Avr. elong (%)
**2.3**	996.1	7.8	925.0	15.0	14.1	0.6
**2.12**	1016.7	1.4	950.0	5.0	13.8	0.2
**2.11**	1023.6	4.1	965.0	5.0	13.3	1.0
**2.29**	1032.0	9.1	970.0	5.0	8.9	2.3
**2.30**	993.3	16.4	941.7	2.9	9.6	0.8
AHT condition						
**2.3 AHT**	745.8	3.7	697.7	3.7	18.3	1.1
**2.12 AHT**	746.5	3.0	697.8	2.2	17.0	2.6
**2.11 AHT**	741.3	7.1	696.1	5.1	18.4	1.3
**2.29 AHT**	739.8	16.6	705.3	0.9	15.1	1.5
**2.30 AHT**	744.4	4.3	694.8	4.1	11.1	0.9
Parent material						
**21NiCrMo2**	614.0	0.9	424.0	3.7	22.4	1.0

**Table 7 materials-15-08972-t007:** HV1 hardness measurements of the parent material and PBF-LB/M (as-built and AHT) material.

Parameter Group	HV1 (XY)	SD	HV1 (YZ)	SD
**2.3**	333	12	343	5
**2.11**	346	21	351	6
**2.12**	334	25	345	7
**2.29**	349	10	357	6
**2.30**	331	20	339	5
**2.3 AHT**	263	4	261	4
**2.12 AHT**	272	5	264	3
**2.11 AHT**	266	5	261	2
**2.29 AHT**	267	3	266	4
**2.30 AHT**	259	4	261	3
Parent material				
**21NiCrMo2**	203	4	-	-

## Data Availability

Not applicable.

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
