# Peer review of "Processability of 21NiCrMo2 Steel Using the Laser Powder Bed Fusion: Selection of Process Parameters and Resulting Mechanical Properties"

_materials, 2022, doi:10.3390/ma15248972_

Round 1
Reviewer 1 Report
1. This paper is very well written. It presents an important topic in the current research field of additive manufacturing. Analysing the possible application of 21NiCrMo2 15 low-alloy steel to be used in gears with PBF-LB/M, and allow for obtaining the proper strength properties.
2. Line 41: Finish the sentence with a full stop.
3. The sentence in line 42 and 43 in not clear.
4. Improve the sentence in line 45.
5. In lines 61-63, I suggest that the authors improve the writing ‘that kind of paper’
6. I suggest that the author add a flowchart in section 2 to explain the methodology to increase clarity of the work performed.
7. The abstract and introduction is well written, contains all the necessary information, and has a good flow.
8. In Figure 1, indicate the powder grains and satellites.
9. Please explain why the 650 celcius was chosen for heat treatment in the furnace.
10. How many samples were used in the experiment? The average values were taken from the sample results?
11. Improve the sentence in lin 283
12. In Table 4, why is the hatching distance for the extreme minimum presented with 2 decimal points, whereas for the deterministically estimated group, with 4 decimal points? I suggest the same decimal number points for consistency.
13. In Table 5, the coefficients presented are with different decimal number points, please amend the table and consistency and with the required decimal point necessary.
14. In the results, were any comparisons made with findings from the literature to ensure results validity?
15. Figure 5: Please indicate the red boxes in Figure 5 (b)
16. The quality of images in Figure 6 in low, please enlarge to increase clarity of the figures.
17. In Table B1, some columns used point and decimals, whereas in the other column, commas were used. Please change all to points formats for decimals.
18. The paper contains 33 references, from which 21 are from the last 5 years, which show the importance of the topic and that a thorough literature review was conducted.
Reviewer 2 Report
This paper studies engineering applicability of 21NiCrMo2 steel processed by laser bed fusion additive manufacturing. Firstly, a numerical method using the quadratic regression is proposed to find precise process conditions, laser power, scanning velocity and hatching distance, for porosity fraction to be around 0.3%. Secondary, this numerical method is confirmed by comparing predicted porosity faction with value measured by experimental microstructure. Finally, mechanical properties, stress-strain curves and harnesses of as-build, heat-treatment and parent material samples are experimentally presented in some process conditions designed by the numerical method.
It is considered that valuable information is well obtained and summarized for engineering field of powder bed fusion additive manufacturing using steel. However, some minor revied points are remained as follows:
(1) The layer thickness is set constant in this study. Then the quadratic regression model uses three process parameters, laser power, scanning velocity and hatching distance. The layer thickness is very important parameter as showed in energy density, Eq.(1). Please add consideration in case of the layer thickness included in the regression model.
(2) In Figure 5 (b), What are the small black grains? Carbide? Pleas add some comment.
(3) There is no definition of the p coefficient in lines 220-222. Please add the p meaning.
(4) In Eq(1), tL is lt ?
